# Skeletal Muscle Health and Cognitive Function: A Narrative Review

**DOI:** 10.3390/ijms22010255

**Published:** 2020-12-29

**Authors:** Sophia X. Sui, Lana J. Williams, Kara L. Holloway-Kew, Natalie K. Hyde, Julie A. Pasco

**Affiliations:** 1IMPACT—The Institute for Mental and Physical Health and Clinical Translation, Deakin University, Barwon Health, Geelong, VIC 3220, Australia; l.williams@deakin.edu.au (L.J.W.); k.holloway@deakin.edu.au (K.L.H.-K.); natalie.hyde@deakin.edu.au (N.K.H.); julie.pasco@deakin.edu.au (J.A.P.); 2Department of Medicine-Western Health, The University of Melbourne, St Albans, VIC 3021, Australia; 3Department of Epidemiology and Preventive Medicine, Monash University, Melbourne, VIC 3181, Australia; 4Barwon Health, University Hospital Geelong, Geelong, VIC 3220, Australia

**Keywords:** skeletal muscle health, sarcopenia, cognitive function, dementia, cognitive decline, cognitive impairment, vitamin D, inflammation, oxidative stress, lifestyle risk factors

## Abstract

Sarcopenia is the loss of skeletal muscle mass and function with advancing age. It involves both complex genetic and modifiable risk factors, such as lack of exercise, malnutrition and reduced neurological drive. Cognitive decline refers to diminished or impaired mental and/or intellectual functioning. Contracting skeletal muscle is a major source of neurotrophic factors, including brain-derived neurotrophic factor, which regulate synapses in the brain. Furthermore, skeletal muscle activity has important immune and redox effects that modify brain function and reduce muscle catabolism. The identification of common risk factors and underlying mechanisms for sarcopenia and cognition may allow the development of targeted interventions that slow or reverse sarcopenia and also certain forms of cognitive decline. However, the links between cognition and skeletal muscle have not been elucidated fully. This review provides a critical appraisal of the literature on the relationship between skeletal muscle health and cognition. The literature suggests that sarcopenia and cognitive decline share pathophysiological pathways. Ageing plays a role in both skeletal muscle deterioration and cognitive decline. Furthermore, lifestyle risk factors, such as physical inactivity, poor diet and smoking, are common to both disorders, so their potential role in the muscle–brain relationship warrants investigation.

Emerging evidence indicates that sarcopenia is associated with an increased likelihood of cognitive impairment and hence the development of dementia. This review of the literature summarises current knowledge about potential links between sarcopenia and dementia, risk factors and underlying biological mechanisms.

## 1. Dementia

Dementia is a condition that encompasses many progressive and acquired neurocognitive disorders [1]. Its core feature is a loss of intellectual abilities severe enough to disturb social and occupational functioning. Dementia affects multiple cognitive domains, such as executive function, complex attention, learning and memory, language and perceptual motor and social cognition [2]. Worldwide, approximately 50 million people have dementia, and this number is projected to reach 82 million by 2030; an estimated 5–8% of people aged 60+ years have dementia [3].

Alzheimer’s disease (AD) is a cause of dementia, accounting for 50–70% of dementia cases. AD is a progressive, degenerative disorder involving loss of memory, thinking and language skills and behavioural changes [1]. Globally, nearly 44 million people were affected by Alzheimer’s disease in 2016 [4]. One in nine people aged 65+ years has AD, and its prevalence increases with age. In 2019, 3% of people aged 65–74 years had AD, 17% of people aged 75–84 years and 32% of those aged 85+ years [4].

Mild cognitive impairment (MCI) is a pre-dementia stage—a condition between normal, age-related cognitive decline and dementia [5]. Cognitive impairment, or cognitive decline, is associated with increased mortality and substantially reduces quality of life [6,7,8]. The prevalence of cognitive impairment varies depending on the criteria and participants’ demographic characteristics; however, it increases with age [9,10] and appears to be higher in men [11].

People with MCI are usually able to perform normal activities and compensate for small changes with minimal difficulty. MCI is categorised either as an amnestic MCI subtype in which memory is impaired or as a non-amnestic MCI subtype [6]. Most studies measure the prevalence of MCI in the general population as 16–20% [6]. However, its prevalence varies considerably with the criteria, definitions and assessment instruments applied [6]. The few existing measurements of MCI incidence rates range from 5 to 168 per 1000 person-years [6]. Impaired cognitive function predicts dementia in later life [6,12], and multi-domain MCI confers a higher risk of progressing to dementia than single-domain MCI [6]. There are currently no pharmacological or psychological therapies for dementia [13]; however, MCI is potentially recoverable, and sufferers do not necessarily develop dementia [2]. Intervention at the MCI stage may enable modification of the trajectory towards dementia [2].

## 2. Sarcopenia

Sarcopenia is the age-related deterioration of skeletal muscle and a key component of physical frailty, including poor physical function and muscle strength. Prevalence estimates vary from 9.9% to 40.4% due to differences in demographics across geographical areas [14,15] and also to the use of different criteria for identifying cases. Moreover, the application of different case definitions to the same population can cause sarcopenia prevalence estimates to vary by 40% [14,16,17,18,19,20,21]. Data from the Geelong Osteoporosis Study (GOS) recently described the consequences of applying different criteria and cut-off points in a homogeneous sample to assess prevalence estimates for the Australian population [22]. The main finding was that, across several definitions, the prevalence of sarcopenia increased with increasing age; however, the varied criteria and cut-off points resulted in inconsistent findings regarding case ascertainment for sarcopenia. The varied levels of agreement between the definitions highlight the need to carefully interpret studies that report sarcopenia prevalence in the context of the population studied and the criteria employed.

Sarcopenia components such as muscle strength and mass decline with age. The average decline from peak muscle strength by age 40 years is 16.0%, and 40.9% for people aged 60+ years [23]. Muscle mass decreases by approximately 3–8% per decade after 30 years of age and accelerates after age 60 years [24]. There appears to be a sex difference in the rate of decline in muscle mass and strength, with women experiencing a greater rate of decline after menopause [25]. Information from the GOS recently reported the first normative data for handgrip strength and muscle quality and described the relationship with age, anthropometry and measures of body composition in women [26]. This population-based study contributes to the evidence base by providing geographic-specific data for assessing muscle weakness in conditions such as sarcopenia and frailty. Similar data are needed for men.

Some studies that have investigated the overlap between frailty and sarcopenia have demonstrated that people with frailty are likely to have sarcopenia, but not all people with sarcopenia are frail [27]. For example, a cross-sectional study in Singapore examined participants attending an outpatient clinic where frailty was assessed using the Edmonton Frail Scale [28] and sarcopenia was assessed using the rapid five item (SARC-F) questionnaire [29]. Of 115 patients aged 65+ years, 44.3% of patients had sarcopenia, 31 (27.0%) were frail, and 27 (23.5%) were both frail and sarcopenic. Of those with frailty, 87.1% also had sarcopenia, whereas 47.1% of sarcopenic patients met the criteria for frailty. A study [30] in the Netherlands included data from 227 participants (aged 65+ years) in community care settings. Sarcopenia was identified using the algorithm from the European Working Group on Sarcopenia in Older People 2010, while physical frailty was assessed by using the Fried criteria and the International Association of Nutrition and Aging proposed frailty scale. Sarcopenia was identified in 23.3% and physical frailty in 8.4–9.3% of the participants. The risk of having sarcopenia increased with age in those without frailty; however, the older people with frailty were 60% more likely to have sarcopenia. It should be noted that the criteria for identifying sarcopenia and frailty were applied differently; however, the findings were consistent.

## 3. Sarcopenia and Cognitive Function

While the literature describes bi-directional associations between physical and mental decline, little is known about skeletal muscle deficits as risk factors for poor cognitive performance. Although results are equivocal, emerging evidence indicates that sarcopenia is associated with an increased likelihood of cognitive impairment. A systematic review and meta-analysis found that sarcopenia was associated with cognitive dysfunction (adjusted odds ratio 2.2, 95% CI 1.2–4.2) [31]. This association was not modified by ethnicity, sex or assessment instruments [31]. However, this meta-analysis only located cross-sectional studies [31]; prospective studies are warranted to identify potential causal relationships. A recent cross-sectional study [32] examined the association between sarcopenia and cognitive impairment in 201 older Korean women. Cognitive function was assessed using the Mini-Mental State Examination (MMSE), while sarcopenia was identified based on the Asian Working Group for Sarcopenia’s definition. This study reported that women with pre-sarcopenia and sarcopenia were two and five times more likely to have cognitive impairment, respectively, than women who were non-sarcopenic. In contrast, a French study involving 3025 women aged 75+ years, which examined associations between cognitive impairment and six operative sarcopenia definitions [33], found that sarcopenia was not associated with cognitive impairment, regardless of the definition applied. Furthermore, researchers in the United States of America (USA) reported that sarcopenia was not associated with cognitive functioning in adults aged 60–69 years, but it was for those aged 70+ years [34]. These contradictory results may be due to the components of sarcopenia assessed, which might play different roles in linking physical and mental decline.

The components of sarcopenia, especially muscle strength and gait speed, are well documented as being associated with cognitive function. However, how muscle mass contributes independently of the other components of sarcopenia remains unclear. Muscle mass is associated with muscle strength, possibly in a non-linear way [33], but the latter may be a better predictor of cognitive decline [35]. For example, a cross-sectional study of 223 US adults aged 40+ (mean age 68.1 years, 35% male), drawn from the general community, examined the effect of sarcopenia on physical and cognitive functioning [36]. It measured muscle strength using a handheld dynamometer and lean mass (a surrogate measure of muscle mass) using bioelectrical impedance analysis (BIA); participants exhibiting low lean mass were classified as pre-sarcopenic, those with low lean mass and muscle strength were considered sarcopenic, and those with high lean mass and low muscle strength were categorised as non-sarcopenic [36]. The Montreal Cognitive Assessment was used to assess visuospatial function, executive function, attention, language, memory and orientation; the Ascertaining Dementia questionnaire was used to identify changes in memory and problem-solving ability [36]. Individuals with sarcopenia were six times more likely to have cognitive impairment than the non-sarcopenic group (healthy controls) [36]. After adjusting for confounders, there was a threefold greater risk of cognitive impairment for the sarcopenia group [36]. Thus, muscle strength, rather than muscle mass, appeared to drive the relationship between sarcopenia and cognitive impairment, suggesting that interventions designed to improve muscle strength may also reduce cognitive decline in middle-aged and elderly people [36]. This study was limited by its cross-sectional design and measurement of lean mass without adjusting for body mass index [36]. Data from the GOS recently investigated components of sarcopenia individually in relation to cognitive function [37]. The main finding was that muscle strength and gait speed, rather than muscle mass, are better indicators of poor cognitive function, especially in the domains of information processing (psychomotor function), visual attention and overall performance, even after accounting for differences in age, education status and physical activity.

Further research is warranted to identify why studies have produced conflicting conclusions, which components of sarcopenia are associated with cognitive impairment and whether they are risk factors for cognitive decline. The following sections present a critical appraisal of the evidence for the relationship between muscle mass and function (components of sarcopenia) and cognitive function.

## 4. Muscle Mass and Cognitive Function

In a cross-sectional study, Nourbashemi and colleagues (2002) [38] examined whether lean mass was associated with cognitive deficits among 7105 women. Lean mass was measured using dual-energy X-ray absorptiometry (DXA) and general cognitive function using the short portable mental status questionnaire (SPMSQ). The authors found that women in the lowest quartile of lean mass had a 1.43-times higher risk of general cognitive impairment than those in the highest quartile of lean mass. However, the authors pointed out that this study had several limitations, notably using SPSMQ to screen for cognitive impairment without clinical assessment of dementia, not assessing potential confounders (e.g., physical activity) and its single-sex sample [38]. Similarly, in a German cross-sectional study of the relationship between cognitive function, body composition and nutrition, involving 4095 hospitalised patients (71.3% female), cognitive function was evaluated using MMSE and lean mass using BIA [39]. Results showed that a 5.9% loss of lean mass was associated with an increase in score from 2.1 to 3.0, indicating cognitive deterioration. A cross-sectional study of 51 healthy and community-dwelling older men in the United Kingdom (UK) reported conflicting results [40]. The study measured muscle volume in the neck area [41], general cognitive function using MMSE and cognitive domains of memory and executive function using Rey’s auditory–verbal declarative memory test and the controlled word association test, respectively [40]. It is noteworthy that muscle volume is rarely reported in the literature and is different from muscle mass. An estimate of prior general cognitive ability was also assessed using Benton’s visual retention test (a test for visual memory) and the national adult reading test. This study found no association between neck muscle volume and cognitive abilities [40] and found that total muscle volume was negatively associated with estimated prior cognitive ability [40]. The results suggested that individuals with lower cognitive abilities are more likely to have larger muscle size as they age [40]. The authors speculated that this finding may have been due to men with lower cognitive function being more likely to engage in manual work.

In a prospective study of sarcopenia as a risk factor for cognitive impairment, 297 participants aged 65+ years without cognitive impairment at baseline were followed over a period of five years [42]. Mean lean mass did not differ between groups (normal cognitive function, MCI and dementia); thus, no significant association between lean mass and the risk of developing cognitive impairment was detected. However, the authors suggested that this lack of association may have been due to insufficient statistical power [40,42].

There may be sex differences in the association between muscle mass and cognitive function, as muscle mass varies substantially between men and women [33]. However, no evidence has confirmed this hypothesis. For example, in a recent longitudinal study in Hong Kong, 2737 cognitively healthy men and women aged 65+ years were followed over four years [43]. Lean mass was measured using DXA and general cognitive function using MMSE. Lower lean mass was associated with a higher risk of general cognitive decline in men; however, this association was not sustained after adjusting for confounders, and no relationship between lean mass and general cognitive decline was found in women. The sample was not randomly selected, which could have led to a bias towards healthier older adults, both physically and cognitively.

Intervention studies of muscle mass and changes in cognitive function have produced unconvincing results. For example, Lauque et al., 2004 [44] conducted a randomised trial of the influence of oral nutritional supplements on physical and mental status, including body composition and cognitive function, in patients with AD aged 65+ years, over three months. Forty-six patients were treated with nutritional supplements for three months and 45 received their usual care; lean mass increased in the nutrition supplement group, but no change in cognitive function was detected.

Some studies have examined body composition in patients with AD and dementia. A cross-sectional study of US adults aged 60+ years identified as cognitively normal (*n* = 70) or with early-stage AD (*n* = 70) had body composition measured by DXA and cognitive function by a standardised psychometric battery [45]. Lean mass was lower in the patients with AD, after controlling for sex [45]. In contrast, in another cross-sectional study involving 1462 women aged 75 years or older in France, in which lean mass was again measured using DXA but cognitive function was evaluated by SPMSQ or MMSE, lean mass was not associated with dementia [46]. These cross-sectional studies demonstrate the limited evidence for an association between muscle mass, cognitive impairment and dementia.

## 5. Muscle Strength and Cognitive Function

Cross-sectional studies have consistently demonstrated an association between muscle strength and cognitive function. A study of 3025 French women aged 75+ years, recruited from the community, measured general cognitive function by SPMSQ and muscle strength by handgrip strength (HGS) [33]. Lower HGS was associated with cognitive impairment (OR 1.81, 95% CI 1.33–2.46) [33]. Similarly, in a study of Japanese adults aged 85 years (90 men, 117 women), recruited from the community, those with higher MMSE scores had greater mean HGS in both the right hand (21.8 ± 7.1 vs. 19.3 ± 5.8 kg, *p* = 0.009) and the left (20.6 ± 6.7 vs. 17.9 ± 5.5 kg, *p* = 0.003) and greater isometric leg extensor strength (22.7 ± 8.7 vs. 20.7 ± 9.1 kg, *p* = 0.18) [47]. This association persisted after adjustment for confounders [47]. These two studies, however, included mainly cognitively normal individuals, suggesting possible sample bias. Moreover, while the studies mentioned above suggest a positive association between muscle strength and general cognitive function, they offer no insight as to which domains of cognitive function are most closely related to muscle strength.

The association between muscle strength and specific cognitive domains has been examined using neuropsychological batteries. For instance, in a study involving 1799 US participants aged 60+ years, knee extensor isokinetic strength was measured using a kinetic communicator isokinetic dynamometer and visual spatial and motor speed processing using the digit symbol substitution test (DSST) [48]. The participants were divided into four groups by quartiles of quadriceps strength. DSST scores were greater in higher quadriceps strength groups, indicating that muscle strength is associated with speed of processing and visual–spatial processing [48]. In a study in the Netherlands of 555 participants at ages 85 (35% men) and 89 years (29% men), the adjusted scores of cognitive tests were categorised into tertiles; HGS was associated with scores in tests for processing speed and memory for both age groups, but not with attention at age 89 years [49]. These studies suggest that muscle strength is associated with certain cognitive domains and less related to other domains in older adults. However, specific cognitive domains are described poorly in the current literature.

While cross-sectional studies support an association between muscle strength and cognitive function in general and specific domains, some longitudinal studies indicate a bi-directional association between poor muscle strength and poor cognitive function. In a study of 2160 non-institutionalised Mexican Americans (57.5% women) aged 65+ years, HGS at baseline was associated with greater cognitive decline (MMSE; β estimate = 1.28, se = 0.16; *p* = 0.0001) over a period of six years [50]. In 2381 Mexican American men and women without disabilities aged 65+ years, a decline in HGS was observed over seven years for participants with poor global cognitive function (measured by MMSE) compared to those with good cognitive function [51]. The Women’s Health Initiative Memory Study (WHIMS) in the US, involving 1793 women aged 65–80 years, measured reciprocal changes in general cognitive function (MMSE scores) and HGS over six years [52]. Another longitudinal study reported associations between reduced HGS decline and better attention, memory and processing speed at baseline; however, lower baseline HGS was not associated with an accelerated decline in the measured cognitive domains [49]. Further longitudinal studies are required to determine if a bi-directional relationship exists.

The relationship between muscle strength and cognitive function has been examined in intervention studies. A randomised study in Austria assessed the effect of structured strength training on cognitive function in 42 men and women (mean age 86.8 years) with cognitive impairment and frailty and found that muscle strength in the muscle training group increased over 10 weeks compared to the control group [53]. Even though a linear relationship was observed between muscle strength and MMSE scores in the muscle training group, the mean MMSE scores of the training and control groups did not differ [53]. The short intervention period may have contributed to this finding. Other studies have investigated whether muscle strength intervention improves cognitive function in specific domains. A Brazilian study involved 62 older adults aged 65–75 years undertaking 24-week resistance training at two intensities to examine the impact of muscle strength training on cognitive function [54]. Participants were randomly assigned to control, experimental moderate- and experimental high-intensity training groups [54]. Cognitive function was tested using the Wechsler adult intelligence scale, third edition (WAIS-III), to assess specific cognitive domains, such as central executive, short-term memory and long-term memory [54]. The training groups showed improvement on neuropsychological tests, such as the forward digit span and immediate recall tests, indicating that the intervention improved cognitive function [54]. Similarly, Berryman et al. studied the effect of eight weeks of aerobic strength training on the executive function of 47 healthy adults (mean age 70.7 ± 5.6 years) and reported increased muscle strength and improved executive function [55]. These studies indicate that improving muscle strength improves cognitive function, at least in some domains.

While general cognitive function has been assessed using global cognitive tests, such as MMSE and its versions, these tests have little ability to identify subtle differences within healthy populations [56]. Further research must incorporate measures of general and specific cognitive function to clarify the relationships between muscle strength and cognitive function.

A cross-sectional study involving 1038 Korean men and women with severe cognitive impairment, aged 65+ years, reported that each 8-kg decrease in HGS was associated with a 59% increased likelihood of dementia (adjusted OR 1.59; 95% CI 1.19–2.14) [57]. Similarly, the Religious Orders Study, involving 877 US men and women without dementia, reported that each 1 kg deficit in baseline HGS conferred a 1.5% greater risk of developing AD over 5.7 years (adjusted hazard ratio 0.986; 95% CI 0.973–0.998) [58].

## 6. Physical Performance and Cognitive Function

Gait speed has been associated with health outcomes and lifespan [59] and is considered to indicate higher-level cognitive functioning, because it is involved in complicated cognitive functions such as attention, memory, motor function and perception. However, most studies perceive gait performance and cognitive function as independent constructs. Several cross-sectional studies suggest that gait speed is associated with cognitive function. A study of 4000 Chinese men and women from the community examined the association between cognitive function and physical performance, using a 6-m walk speed test and chair stand test. These authors found that the cognitive impairment group had poorer performance in gait speed tests than the non-cognitively impaired control group (0.89 ± 0.024 vs. 1.02 ± 0.004 m/s in men and 0.85 ± 0.009 vs. 0.93 ± 0.005 m/s in women, both *p* < 0.001) and chair stand tests (13.99 ± 0.05 s vs. 12.57 ± 0.09 s in men and 14.45 ± 0.27 s vs. 13.07 ± 0.12 s in women, both *p* < 0.001) [60]. Additionally, a US study involving 44 men and women with amnestic MCI (mean age = 79.3 ± 4.7 years), 62 with non-amnestic MCI (mean age 81.8 ± 6.2 years) and 295 healthy individuals (mean age 81.8 ± 6.2 years) compared gait performance across the groups [61]. Tests of gait performance were conducted using computer-based analyses of gait ability that included pace, rhythm and variability. Cognitive function was assessed using a neuropsychological test battery for general cognitive function and specific cognitive domains, including memory, executive function, attention and language [61]. The results showed that gait, even if measured in different ways, was worse in participants with MCI than in the controls.

Gait speed has been associated with specific cognitive domains, such as executive function, visuospatial ability and psychomotor function [62], and poor gait performance with executive function impairment. For example, Coppin et al., 2006 [63] reported slower gait speed in participants with poor executive function than those with high executive function. Several specific cognitive domains were not associated with gait in other studies. For instance, Martin et al. [62] reported that gait measures were not associated with memory, perhaps due to test sensitivity or lack of study power, but possibly because gait is only associated with specific domains of cognitive function [64].

Some longitudinal studies show that reduced gait speed in ageing predicts declines in general cognitive function and several specific domains. A US study of 204 healthy older adults (58% female) found that gait speed declined by 0.02 m/s/year for up to 12 years prior to the onset of MCI, as assessed using standardised neurologic examinations [65]. Similarly, in the Health, Ageing and Body Composition (Health ABC) Study, involving 2776 men and women aged 75–85 years, DSST scores underwent the largest declines in participants in the lowest quartile of gait speed, indicating that gait speed predicts decline in attention and psychomotor speed in the elderly [66].

Conversely, several studies have demonstrated that general and specific cognitive decline predict gait speed decline. In the Health ABC Study, assessment of 2349 men and women (mean age 75.6 years) over three years showed that lower global cognitive function and executive function were associated with greater gait speed decline [67]. Executive cognitive deficits during ageing were found to account for gait slowing [68]. Lower general cognitive function, verbal memory and executive function were associated with greater gait speed decline each year [69]. More specifically, Soumare et al., 2009 reported that poorer verbal fluency and slower psychomotor speed were associated with larger declines in gait speed [70].

Several studies have investigated whether fast gait speed predicts cognitive decline. In an Italian longitudinal and population-based study involving 660 older adults aged 65+ years, followed over three years, usual gait speed, fast speed and speed during “walking while talking” were measured at baseline and follow-up; cognitive function was assessed using MMSE at baseline and follow-up and adjusted for confounders [71]. Only fast gait speed predicted general cognitive decline [71]. Several other studies have examined bi-directional associations between muscle function and cognitive function. For example, change in general cognitive function was associated with change in physical performance, including gait speed; however, baseline physical performance, including gait speed, was not associated with cognitive change [52].

A 2004 systematic review included seven studies comparing gait in dementia and healthy older adults and reported that the former had shortened step length and lower walking speed [72]. A more recent review of the association between gait and cognitive function in epidemiological and neuropsychological studies of gait disorders and dementia revealed that gait disorders were associated with moderate and severe dementia, but not mild dementia [73].

## 7. Muscle Quality, Muscle Density and Cognitive Function

Muscle quality is defined as the force generated by each volumetric unit of muscle tissue and may reflect the amount of contractile protein, fat infiltration (myosteatosis), aerobic capacity and other physiological properties of the muscle [74,75]. However, as there is no standardised protocol for quantifying muscle quality, few studies have examined low muscle quality in association with cognitive function. One US study examined the relationship between muscle quality (isokinetic strength in relation to leg muscle mass) and cognitive function in adults aged 60+ years, finding a positive relationship in both sexes [76].

Bone mineral density and lean mass are correlated [77]. A UK study investigated cognitive impairment and bone using peripheral quantitative computed tomography (pQCT), and reported that gait speed, but not bone density/structure, was associated with MCI [78]. A similar Italian study suggested that low bone mineral density is an early marker of cognitive decline in the elderly [79]. These studies used MMSE to screen for cognitive impairment, which examines global cognitive function.

Data from the GOS [80] recently examined pQCT-derived muscle density (a surrogate measure of muscle quality) and specific domains of cognitive function. This study included 281 men (aged 60–95 years) whose radial and tibial muscle density were measured using pQCT and body fat and appendicular lean mass were measured using DXA. Cognitive function was assessed in different domains using CogState Brief Battery. Regression analyses revealed that muscle density was associated with psychomotor function and visual learning, independent of education and physical activity. While there was some evidence that serum Tumour Necrosis Factor alpha (TNF-α) attenuated the association between radial muscle density and psychomotor function, there was little other evidence that inflammation and adiposity explained the association between muscle density and cognitive function, at least with the parameters investigated. The age-related associations did not occur simultaneously between muscle density and all cognitive domains that were tested.

## 8. Potential Mechanisms

### 8.1. Vitamin D

Circulating vitamin D helps to sustain both skeletal muscle and brain function. A systematic literature review and meta-analysis of 13 randomised controlled studies of the effect of supplemental vitamin D on muscle strength, gait and balance in older adults concluded that it improves muscle strength and balance [81]. There is likely to be no ideal level for supplemental vitamin D, which will depend on the background vitamin D status, from dietary sources and exposure to ultraviolet radiation (UV), as a threshold effect is likely. Vitamin D deficiency also plays a role in brain function deficits. Another systematic literature review and meta-analysis (37 studies) found that low vitamin D is associated with poor cognitive function and a higher risk of developing dementia [82]. More research is needed to determine if low vitamin D concurrently influences both sarcopenia and cognitive decline.

### 8.2. Inflammation and Oxidative Stress

Fat infiltration into skeletal muscle occurs with ageing and is a characteristic of sarcopenic obesity, whereby sarcopenia exists in the face of excessive body fat accumulation. As obesity is an inflammatory state [83,84] and inflammation has detrimental effects on both muscle [85] and brain [76], it is plausible that inflammation, or indeed adiposity itself, might play a role in linking low muscle density with poor cognitive function.

Systemic, chronic, low-grade inflammation in ageing, termed inflamm-ageing [86], is a major risk factor for both mental and physical disease [87,88]. Inflamm-ageing contributes to changes in skeletal muscle properties [89] and sarcopenia [85] and mediates AD and cognitive deficits in the elderly [85,90]. Biomarkers of inflammation, such as interleukin 6 (IL-6) and TNF-α, have been associated with muscle atrophy in both human and animal studies [85,91,92] and age-related cognitive decline in humans [90,93,94,95].

Considerable evidence demonstrates a strong association between inflammation and risk of cognitive decline. For example, a cross-sectional US study involving 269 participants (mean age 67 years) found that IL-6 was negatively associated with MMSE [96]. Cross-sectional studies in Germany (*n* = 369) [97] and Italy (*n* = 744) [98], involving participants aged 65+ years, found negative associations between inflammatory markers (C-reactive protein, CRP and IL-6) and performance in specific cognitive domains. The association between inflammation and cognitive function has been confirmed through longitudinal studies. In the Health ABC study of black and white men and women aged 70–79 years over two years (*n* = 3013) and eight years (*n* = 2509) [99], general cognitive function was measured using a modified MMSE. Individuals with higher levels of IL-6 and CRP had a 24% increased risk of cognitive decline at two-year follow-up and similar results at eight years [99,100]. Other longitudinal studies in the Netherlands found that CRP and IL-6 were negatively associated with memory [101,102], learning [102], attention [101], cognitive speed [101] and language [102]. A meta-analysis that included 1098 patients with AD and 1094 controls in 27 studies reported oxidative damage in peripheral blood vessels during early-stage AD [103].

A US cross-sectional study of the pathophysiological links between sarcopenia and dementia involved 445 women and 442 men aged 60+ years [76]. Muscle quality was assessed using isokinetic strength per unit muscle mass, and cognitive function using the WAIS-III digit symbol [76]. This study found that high-sensitivity CRP was associated with both cognitive impairment and poor muscle quality in women [76].

### 8.3. Vitamin D, Exercise and Inflammation

Vitamin D inhibits inflammation through regulating the production of inflammatory cytokines and inhibiting the proliferation of proinflammatory cells [104], which could be beneficial to skeletal muscle heath and brain health [105]. A Iranian study found that vitamin D_3_ supplementation reduced the mRNA expression levels of IL-17A and IL-6, but increased the level of IL-10 [106]. This study focused on patients with multiple sclerosis.

Exercise through muscle contractions induces myokines (e.g., IL-6, brain-derived neurotrophic factor, BDNF). Recovery from IL-6 peak following exercise can dampen inflammation and oxidative burst activity; the anti-inflammatory effect of BDNF may contribute to this recovery. It is well known that chronic exercise downregulates systemic inflammation and is an effective management strategy for insulin resistance [107]. Thus, exercise-induced downregulation of IL-6 and the anti-inflammatory effect of BDNF may be related pathways. Thus, dietary, UV-associated or supplementary vitamin D combined with exercise may affect both brain and skeletal muscle health. For example, a study in the USA suggested that supplemental vitamin D combined with intense exercise may enhance the recovery of muscle strength in adults who suffer a muscle damage event [108].

## 9. Common Lifestyle Risk Factors

Lifestyle factors are likely to mediate the association between sarcopenia and cognitive impairment. Physical inactivity, poor diet and smoking are examples of poor health behaviours common to both disorders.

### 9.1. Physical Inactivity

A 2014 systematic review of 47 cohort studies highlighted that physical activity was associated with various cognitive outcomes [109]. Over 87% of the studies and 100% of the cross-sectional studies demonstrated a relationship between physical inactivity and cognitive impairment, including AD [109]. Resistance training is a recognised means of improving skeletal muscle mass and function [110]. Contracting skeletal muscles produce cytokines [111], which affect lipid and glucose metabolism.

### 9.2. Poor Diet

Adequate nutrition is essential for maintaining skeletal muscle, and sarcopenia is associated with inadequate caloric and protein intakes. Prospective studies describe the protective effect of dietary protein for sarcopenia [112] and positive associations between protein intake and muscle mass [113]. Furthermore, muscle mass losses are considerably reduced in older individuals with high protein intake [114]. Malnutrition is substantially associated with frailty and sarcopenia in hospitalised older patients [115] and in geriatric rehabilitation patients [116]. It is well documented that poor diet promotes cognitive decline, including AD [117,118]. Consumption of fish (a source of long-chain omega-3 fatty acids) slows cognitive decline in older people without AD, but this study does not suggest that it can treat dementia [119]. A US longitudinal study [120] examined the relationship between poor diet and cognitive decline in participants with a mean age of 48 years. Diet quality was assessed using a self-reported questionnaire, and cognitive function with a neurocognitive test battery. Poorer diet was associated with reduced attention and cognitive flexibility, visuospatial ability and perceptual speed.

### 9.3. Smoking

Tobacco smoke generates free radicals, causing lipid peroxidation, oxidation of protein and other tissue damage in smokers [121]. Nicotine may influence the link between sarcopenia and cognitive impairment, but other toxic components of cigarettes cannot be ignored.

A meta-analysis of smoking as an independent risk factor for sarcopenia included 22,515 participants across 12 studies [122]. A fixed effect model was used to estimate sarcopenia risk for men (OR 1.12, 95% CI 1.03–1.21) and a random effect model for women (OR 1.21 95% CI 0.92–1.59). These sex-specific results are difficult to compare because of the different methods employed [122]. Another study found that lifelong cigarette smoking was associated with higher prevalence of sarcopenia in older adults [123].

Reviews of epidemiological and clinical studies provide inconsistent evidence as to whether smoking is a risk factor for cognitive impairment. Many epidemiological studies have found that smoking is negatively associated with AD [124]. An early study found that patients with AD and/or Parkinson’s disease are more likely to be lifelong non-smokers than those without. However, other studies have detected a positive independent association between smoking and AD. Note that a survivor effect may exist in large-scale longitudinal studies: non-smokers may be more likely to suffer from AD simply due to surviving to an age when it is more common.

### 9.4. Alcohol Consumption

Alcohol consumption is associated with body composition changes, including muscle autophagy, as ethanol inhibits protein synthesis in muscles [125]. However, the current evidence is controversial. A meta-analysis that included 214 research articles involving a total of 13,155 participants found no evidence to support alcohol intake as a risk factor for sarcopenia [126]. However, a cross-sectional study conducted by Daskalopoulou et al., 2020 [127] that investigated the risk factors of sarcopenia and sarcopenic obesity in low- and middle-income countries reported that people who consume moderate levels of alcohol (1–14 units per week for women and 1–21 units per week for men) were more likely to have sarcopenic obesity (OR 1.76, 95% CI 1.21–2.57), but not sarcopenia, compared with people who were in the no drinking or heavy drinking group.

Prenatal alcohol exposure is a risk factor for children’s cognitive development [128]. In contrast, a systematic literature review examined 27 cohort studies (published 2007–2018) [129]. Interestingly, this review reported that moderate alcohol consumption was a protective factor for better cognition in women but no difference was found in men. Data from the GOS found that high alcohol consumption (>20 g/day) was associated with greater tibial muscle density but not with any domains of cognitive function. Higher levels of alcohol consumption did not contribute to or explain the relationships between muscle density and cognitive function [80].

## 10. Summary and Conclusions

The main body of text in this review included and critically examined 30 key research articles in terms of cross-sectional studies, longitudinal studies and clinical trials that demonstrated relationships between skeletal muscle and cognitive function (Table 1, Table 2 and Table 3). Current knowledge about the relationship suffers from a lack of data for cognitive function in specific domains. Research to date has almost exclusively assessed associations between overall cognition and skeletal muscle parameters. Previous studies have shown that different methodologies for assessing cognitive performance may contribute to equivocal results; future studies may investigate biomarkers and utilise neuroimaging techniques for screening MCI.

Epidemiological evidence for links between physical and neurological health is needed. The significance of this review is to provide a rationale for conducting further studies and targeted intervention trials, with a focus of improving the physical and cognitive health of the ageing population.

## Figures and Tables

**Table 1 ijms-22-00255-t001:** Summary of studies that investigated association between skeletal muscle mass (or lean mass) and cognitive function.

Author, Year; Country/Region; Follow-Up Period	Participant Characteristics	Muscle Mass Measurements	Cognitive Function Measurements	Results
**Cross-Sectional Studies**				
1. Nourbashemi et al., 2002 [38]; France	7105 community-dwelling women aged 75+ years	DXA (lean mass)	SPMSQ (focus on orientation, memory; using to identify cognitive impairment in this study)	Women in the lowest quartile of lean mass had a 1.43-times higher risk of general cognitive impairment compared with those in the highest quartile of lean mass
2. Wirth et al., 2011 [39]; Germany	4095 (71.3% female); hospitalised patients	BIA (lean mass)	MMSE (general cognition)	5.9% loss of lean mass was associated with an increased score from 2.1 to 3.0, indicating cognitive deterioration
3. Kilgour et al., 2013 [40]; UK	51 community-dwelling older men mean aged 73.8 ± 1.5 years	CT (muscle volume)	MMSE (global cognition); Rey’s auditory–verbal declarative memory test (memory); the controlled word association test (executive function); Benton’s visual retention test; the national adult reading test	No association between neck muscle volume and cognitive abilities; the total muscle volume was negatively associated with estimated prior cognitive ability; individuals with lower cognitive abilities were more likely to have larger muscle size as they aged
4. Burns et al., 2010 [45]; USA	Cognitively normal (*n* = 70) or with early-stage Alzheimer’s disease (*n* = 70); aged 60+ years	DXA (lean mass)	A standardised psychometric battery (Logical Memory, Free and Cued Selective Reminding Task, Boston Naming, Verbal Fluency, Digit Span Forward and Backward, Letter–Number Sequencing, Stroop Color-Word Test and Block Design MMSE (global cognition)	The lean mass was lower in the patients with Alzheimer’s disease, after controlling for sex
5. Abellan van Kan et al., 2012 [46]; France	1462 community-dwelling women aged 75+ years	DXA (lean mass)	SPMSQ or MMSE (general cognition)	Lean mass was not associated with dementia
6. Sui et al., 2020 [37]; Australia	292 men aged 60+ years; population based	DXA (lean mass)	CogState Brief Battery (psychomotor function, visual identification/attention, visual learning and working memory	No association was detected between lean mass and cognitive function
7. Sui et al.,2020 [80]; Australia	281 men aged 60+ years; population based	pQCT (muscle density)	CogState Brief Battery (psychomotor function, visual identification/attention, visual learning and working memory	Muscle density was associated with cognitive function in the psychomotor function and visual learning
**Longitudinal Studies**				
8. Moon et al., 2016 [42]; South Korean; 5 years follow-up	297 community-dwelling men and women without cognitive impairment at baseline; aged 65+ years	DXA (lean mass)	Korean version of the Consortium to Establish a Registry for Alzheimer’s Disease Clinical Assessment Battery; Korean version of the Mini International Neuropsychiatric Interview; International Working Group on MCI; DSM-IV; Final diagnosis of MCI, dementia was determined by a panel of research neuropsychiatrists	Mean lean mass was not different between groups (normal cognitive function, mild cognitive impairment and dementia); thus, no significant associations between lean mass and the risk of developing cognitive impairment were detected
9. Auyeung et al., 2011 [43]; Hong Kong; four years follow-up	2737 cognitively healthy men and women from the community; aged 65+ years	DXA (lean mass)	MMSE (general cognition)	Lower lean mass was associated with a higher risk of general cognitive decline in men; however, this association was not sustained after adjusting for confounders and no relationship between lean mass and general cognitive decline was found in women
**Intervention Studies**				
10. Cassilhas et al., 2007 [54]; Brazil; 24 weeks	62 older adults aged from 65 to 75 years. Participants were randomly assigned to three groups: control, experimental moderate- and experimental high-intensity training (six exercises including chest press, leg press, vertical traction, abdominal crunch, leg curl and lower back)	Whole-body plethysmography (lean mass)	WAIS III (central executive and short-term memory); WSM-R (visual modality of short-term memory); Toulouse–Pieron’s concentration attention test (attention); Rey–Osterrieth complex figure (long-term episodic memory)	The training groups reported improvement in neuropsychological tests, such as the forward digit span and immediate recall tests, indicating that the intervention improved cognitive function
11. Lauque et al., 2004 [44]; France; 3 months	Patients with Alzheimer’s disease aged ≥65 years; 46 patients were treated with nutritional supplements for three months and 45 received their usual care as a control group	DXA (lean mass)	MMSE (general cognition)	Lean mass increased in the nutrition supplement group; however, no change in cognitive function was detected

BIA: bioelectrical impedance analysis; DSM-IV: Diagnostic and Statistical Manual of Mental Disorders, fourth edition; DXA: dual-energy X-ray absorptiometry; MMSE: Mini-Mental State Examination; pQCT: peripheral quantitative computed tomography; SPMSQ: short portable mental status questionnaire.

**Table 2 ijms-22-00255-t002:** Summary of studies that investigated association between skeletal muscle strength and cognitive function.

Author, Year; Country/Region; Follow-Up Period	Participant Characteristics	Muscle Strength Measurements	Cognitive Function Measurements	Results
**Cross-Sectional Studies**				
1. Abellan van Kan et al., 2012 [33]; France	3025 community-dwelling women aged 75+ years	HGS dynamometry	SPMSQ (used to identify cognitive impairment)	Lower HGS was associated with cognitive impairment
2. Takata et al., 2008 [47]; Japan	Community-dwelling participants aged 85 years (90 men, 117 women)	HGS dynamometry	MMSE (global cognition)	Those with higher MMSE scores were more likely to have greater HGS for both the right hand (21.8 ± 7.1 vs. 19.3 ± 5.8 kg, *p* = 0.009) and the left hand (20.6 ± 6.7 vs. 17.9 ± 5.5 kg, *p* = 0.003) and greater isometric leg extensor strength (22.7 ± 8.7 vs. 20.7 ± 9.1 kg, *p* = 0.18). This association persisted after adjustment for confounders
3. Chen et al., 2015 [48]; USA	1799 population-based men and women aged 60+ years	Isokinetic strength dynamometry	DSST (measuring the visuospatial and motor speed of processing, represented a sensitive measure of frontal lobe executive functions)	The DSST scores were greater in higher quadriceps strength groups, indicating that muscle strength was associated with speed of processing and visual–spatial processing
4. Shin et al., 2012 [57]; Korea	1038 men and women aged 65+ years from the community	Sit-to-stand score; HGS dynamometry	Dementia (identified using the Korean version of GMS B3-K; CSID-K; CERAD)	Each 8-kg decrease in HGS was associated with a 59% increased likelihood of dementia (adjusted OR 1.59; 95% CI 1.19–2.14)
5. Sui et al., 2020 [37]; Australia	292 men aged 60+ years; population based	HGS dynamometry	CogState Brief Battery (psychomotor function, visual identification/attention, visual learning and working memory	for every 1 kg increase in handgrip strength, scores for psychomotor function were 0.003 (log10 milliseconds) lower and for overall cognitive function 0.02 (unitless) higher (both indicating better function).
**Longitudinal Studies**				
6. Taekema et al., 2012 [49]; Netherlands	555 population-based participants at all ages, 85 years at base line (35% men) and 89 (29% men) years at follow-up	HGS dynamometry	Neuropsychological test battery (for assessing global cognitive performance, attention, processing speed and memory)	HGS was associated with scores in tests for processing speed and memory for both age groups, but was not associated with attention at age 89 years
7. Buchman et al., 2007 [58]; USA; 5 years follow-up	877 men and women without dementia	HGS dynamometry	Dementia (Mini Mental State Examination; Health Interview Survey)	Each 1-kg deficit in baseline HGS conferred a 1.5% greater risk of developing AD over 5.7 years (adjusted hazard ratio 0.986; 95% CI 0.973–0.998)
8. Alfaro-Acha et al., 2006 [50]; USA; 6 years follow-up	2160 non-institutionalised Mexican Americans (57.5% women) aged 65+ years	HGS dynamometry	MMSE (measuring cognitive decline)	HGS at baseline was associated with greater cognitive decline, as assessed by MMSE (β estimate = 1.28, se = 0.16; *p* = 0.0001) over a period of six years
9. Raji et al., 2005 [51]; USA; 7 years follow-up	2381 Mexican American men and women aged 65+ years, without disabilities	HGS dynamometry	MMSE (measuring cognitive decline)	A decline in HGS was observed over a period of seven years for participants with poor global cognitive function (measured by MMSE) compared with those with good cognitive function
10. Atkinson et al., 2010 [52]; USA; 6 years follow-up	1793 women aged 65–80 years	HGS dynamometry	MMSE (measuring cognitive decline)	Reciprocal changes in general cognitive function (MMSE scores) and HGS over a follow-up period of 6 years
**Intervention Studies**				
11. Dorner et al., 2007 [53]; Austria; 10-week trial	42 long-term care facility residents (men and women, mean age of 86.8 years) with cognitive impairment and frailty; intervention through involved a structured strength and balance training	Increased muscle Strength	MMSE (measuring cognitive decline)	Muscle strength in the muscle training group increased compared with the control group over a period of ten weeks [53]. Even though a linear relationship was observed between increasing muscle strength and improved MMSE scores in the muscle training group, a difference was not detected in mean MMSE scores between the training and control groups
12. Cassilhas et al., 2007 [54]; Brazil; 24 weeks	62 older adults aged from 65 to 75 years. Participants were randomly assigned to three groups: control, experimental moderate- and experimental high-intensity training (six exercises including chest press, leg press, vertical traction, abdominal crunch, leg curl and lower back)	1 RM test	WAIS III (central executive and short-term memory); WSM-R (visual modality of short-term memory); Toulouse–Pieron’s concentration attention test (attention); Rey–Osterrieth complex figure (long-term episodic memory)	The training groups reported improvement in neuropsychological tests, such as the forward digit span and immediate recall tests, indicating that the intervention improved cognitive function
13. Berryman et al., 2014 [55]; eight weeks	47 healthy older adults (mean age 70.7 ± 5.6 years); compared the effects of three interventions: strength training	Isokinetic strength dynamometer	Generation cognition (MMSE); executive functions, memory, processing speed	Intervention increased muscle strength and improved executive function

CERAD: Consortium to Establish a Registry of Alzheimer’s Disease; CSID-K: Community Screening Interview for Dementia; DSM-IV: Diagnostic and Statistical Manual of Mental Disorders, fourth edition; DSST: digit symbol substitution test; GMS B3-K: Korean version of Geriatric Mental State Schedule B3; HGS: handgrip strength; MMSE: Mini-Mental State Examination; SPMSQ: short portable mental status questionnaire; WSM-R: Wechsler Adult Intelligence Scale III (WAIS III); Wechsler Memory Scale-Revised.

**Table 3 ijms-22-00255-t003:** Summary of studies that investigated associations between physical performance and cognitive function.

Author, Year, Country/Region, STUDY Type	Participant Characteristics	Physical Performance Measurements	Cognitive Function Measurements	Results
**Cross-Sectional Studies**				
1. Auyeung et al., 2008 [60]; Hong Kong	4000 Chinese men and women from the community	6-m walk speed test and chair stand test	CSI-D (identifying dementia)	Cognitive impairment group had poorer performance in gait speed tests than the non-cognitively impaired control group (0.89 ± 0.024 vs. 1.02 ± 0.004 m/s in men and 0.85 ± 0.009 vs. 0.93 ± 0.005 m/s in women, both *p* < 0.001) and chair stand tests (13.99 ± 0.05 s vs. 12.57 ± 0.09 s in men and 14.45 ± 0.27 s vs. 13.07 ± 0.12 s in women, both *p* < 0.001)
2. Verghese et al., 2008 [61]; USA	44 men and women with amnestic MCI (mean age = 79.3 ± 4.7 years), 62 with non-amnestic MCI (mean age 81.8 ± 6.2 years) and 295 healthy individuals (mean age 81.8 ± 6.2 years)	Computer-based analyses of gait ability that included pace, rhythm and variability	Blessed Information-Memory-Concentration test (General cognition); FCSRT (verbal memory); DSST, TMT-B, LFT (executive function); TMT-A and Digit Span forwards-attention); Boston Naming Test (language)	Gait was worse in participants with MCI than in the controls
3. Coppin et al., 2006 [63]; USA	37 community-dwelling individuals aged 65+ years	Complex walking tasks; reference walking tasks	TMT (executive function); MMSE (general cognition)	Reported slower gait speed in participants with poor executive function than those with high executive function
4. Martin et al., 2013 [62]; Australia	422 older people aged 60–85 years	GAITRite walkway	COWAT, Category Fluency, Victoria Stroop test, WAIS-III (Executive function/attention); WAIS-III (Processing speed); Rey Complex Figure copy task (Visuospatial ability); Hopkins Verbal Learning Test—revised, generating scores for total immediate recall, delayed recall and recognition memory and a delayed reproduction after 20 min of the Rey Complex Figure (memory)	Gait measures were not associated with memory
5. Sui et al., 2020 [37]; Australia	292 men aged 60+ years; population based	4-m walk speed test	CogState Brief Battery (psychomotor function, visual identification/attention, visual learning and working memory	For every 1 m/s increase in gait speed, scores for psychomotor function were 0.12 lower, attention 0.08 lower and overall cognitive function 0.49 higher (all better function)
**Longitudinal Studies**				
6. Buracchio et al., 2010 [65]; USA; 20 years follow-up	204 healthy older adults (58% female) aged 65+	9.14-m waking test	MMSE; CDR (identifying dementia)	Gait speed declined by 0.02 m/s/year for up to 12 years prior to the onset of MCI, as assessed using standardised neurologic examinations
7. Inzitari et al., 2007 [66]; 5 years follow-up	2776 men and women aged 75–85 years	6-m walking speed test	DSST (attention and psychomotor speed)	Gait speed predicts decline in attention and psychomotor speed in the elderly
8. Atkinson et al., 2007 [67]; 3+ years follow-up	2349 men and women (mean age 75.6 years)	20-m usual walking speed	3MS (general cognition); ECF; CLOX; 1EXIT (Executive function)	Lower global cognitive function and executive function were associated with greater gait speed decline
9. Deshpande et al., 2009 [71]; Italy; three years follow-up	Population-based study involving 660 older adults aged 65+ years	Walking while talking task	MMSE (general cognition)	Only fast gait speed predicted general cognitive decline

EXIT 15: 15-item Executive Interview; TMT: Block Design Test, Trail-Making Test; CDR: Clinical Dementia Rating Scale; CLOX 1: Clock-Drawing Task; CSI-D: Community Screening Instrument of Dementia; COWAT: Controlled Word Association Test; DSST: Digit Symbol Substitution Test; FCSRT: Free and Cued Selective Reminding Test; LFT: Letter Fluency Test; Modified Mini-Mental Status Examination (3MS); Wechsler Adult Intelligence Scale—Third Edition (WAIS-III).

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
