# Peer review of "Skeletal Muscle Health and Cognitive Function: A Narrative Review"

_ijms, 2020, doi:10.3390/ijms22010255_

Round 1

Reviewer 1 Report

This review manuscript contains the wide range of publications about skeletal muscles and cognitive functions. Authors discussed about the association of several aspects of skeletal muscles (muscle mass, muscle strength, gait speed) and cognitive function. The topics in this manuscript are mature enough to review and well-summarized.

Major comment

  1. Authors discussed about a potential mechanism. Androgens increase both muscle mass and muscle strength, and its blood concentration decreases during aging. Is there any study about relationship between androgens and cognitive function?

  1. Authors also discussed about common lifestyle risk factors, physical inactivity, poor diet, and smoking. Alcohol intake is also an important lifestyle factor. If there is some study about alcohol intake and sarcopenia/ cognitive function, it is more useful to add an information about alcohol.

Minor comment

1. In page 8, IL-6 is shown as a biomarker of inflammation, which is a negative factor for cognitive function. In page 9, IL-6 is also shown as a cytokine produced by contracting skeletal muscle. Exercise is a positive factor for cognitive function. These two aspects of IL-6 seem to be contradiction. Do you have a clear explanation?

Author Response

Comments and Suggestions for Authors

This review manuscript contains the wide range of publications about skeletal muscles and cognitive functions. Authors discussed about the association of several aspects of skeletal muscles (muscle mass, muscle strength, gait speed) and cognitive function. The topics in this manuscript are mature enough to review and well-summarized.

Major comment

  • Authors discussed about a potential mechanism. Androgens increase both muscle mass and muscle strength, and its blood concentration decreases during aging. Is there any study about relationship between androgens and cognitive function?

Response: We thank the reviewer for this comment. However, after a search of the PubMed literature, there is insufficient evidence to demonstrate whether or not androgens play a role in the relationship between skeletal muscle and cognitive function.

  • Authors also discussed about common lifestyle risk factors, physical inactivity, poor diet, and smoking. Alcohol intake is also an important lifestyle factor. If there is some study about alcohol intake and sarcopenia/ cognitive function, it is more useful to add an information about alcohol.

Response: a paragraph has been added to the text (page 21, section 9.4)

Alcohol consumption

Alcohol consumption is associated with body composition changes, including muscle autophagy, as ethanol inhibits protein synthesis in muscles 126. However, the current evidence is controversial. A meta-analysis that included 214 research articles involving a total of 13155 participants found no evidence to support alcohol intake as a risk factor for sarcopenia 127. However, a cross-sectional study conducted by Daskalopoulou et al (2020) 128 that investigated the risk factors of sarcopenia and sarcopenic obesity in low and middle income countries reported that people who consume moderate levels of alcohol (1–14 units per week for women and 1–21 units per week for men) were more likely to have sarcopenic obesity (but not sarcopenia) (OR 1.76, 95%CI 1.21–2.57), compared with people who are in the no drinking or heavy drinking group.

Prenatal alcohol exposure is a risk factor for children’s cognitive development 129. In contrast, a systematic literature review examined 27 cohort studies (published 2007-2018) 130. Interestingly, this review reported that moderate alcohol consumption was a protective factor for better cognition in women but no difference was found in men. Data from the GOS found that high alcohol consumption (>20 g/day) was associated with greater tibial muscle density but not with any domains of cognitive function. Higher levels of alcohol consumption did not contribute to or explain the relationships between muscle density and cognitive function 80.

Minor comment

In page 8, IL-6 is shown as a biomarker of inflammation, which is a negative factor for cognitive function. In page 9, IL-6 is also shown as a cytokine produced by contracting skeletal muscle. Exercise is a positive factor for cognitive function. These two aspects of IL-6 seem to be contradiction. Do you have a clear explanation?

Response: To reduce the confusion, IL-6 has been removed from the manuscript. However, we have added a section to clarify the text in response to this comment.

Modification to the text in the section 8.3 (paragraph 2):

Exercise through muscle contractions induces myokines (e.g. IL-6, brain-derived neurotrophic factor, BDNF) that affect lipid and glucose metabolism. Recovery from IL-6 peak following exercise can dampen inflammation and oxidative burst activity; the anti-inflammatory effect of BDNF may contribute to this recovery. It is well known that chronic exercise down-regulates systemic inflammation and is an effective management strategy for insulin resistance 108. Thus, exercise-induced down-regulation of IL-6 and the anti-inflammatory effect of BDNF may be related pathways.

Reviewer 2 Report

This review study is significantly important to understand previews research outcomes focused on the relation between skeletal muscle and cognitive function and it is a valuable resource for designing future research. However, in my opinion for having a comprehensive and critical research paper, following points might be considered;

Major Comments:

  1. The body of article is well summarized and categorized. But it might be better if the reviewed studies come in form of table, outlining the research methods, outcomes and examined cognitive function domains.
  2. As it is mentioned in the body of article, one of the major reasons of having different results in previous studies, is the cognitive assessment methodology. Therefore, “Summary and conclusion” section should contain critical explanation about differences between cognitive assessments (e.g., MMSE, SPMSQ, WAIS-III &…) and its possible effects on results of study.
  3. Since the section 8 is named as “Potential mechanisms”, some scientific mechanism of action must be outlined in this section. So far, there are number of studies which investigated the possible mechanism of Vitamin D, Inflammation and exercise on cognition and skeletal muscle.
  4. “Summary and conclusion” section might better to contain total number of research articles, average length and other characteristics of studies which are reviewed in this stud

Minor comments:

  1. A comma is needed following the word “neck area” in Page 4, line 163;

“The study measured muscle volume in the neck area general cognitive function using MMSE, and cognitive domains of memory and executive function using Rey’s auditory-verbal declarative memory test and the controlled word association test, respectively.”

2. pQCT is a test for examining bone density. Thus, for better understanding the text, it’s better to change the place of “bone” and “cognitive impairment” in the following sentence. Otherwise it might be misunderstood as a cognitive impairment test [Page 7, line 335];

“A UK study investigated bone and cognitive impairment using peripheral quantitative computed tomography (pQCT), and reported that gait speed, but not bone density/structure, was associated with MCI.”

3. Page 7, line 343; “dual-energy X-ray absorptiometry” is should be written as “DXA” which the abbreviation is mentioned before in Page 4, line 152.

Author Response

This review study is significantly important to understand previews research outcomes focused on the relation between skeletal muscle and cognitive function and it is a valuable resource for designing future research. However, in my opinion for having a comprehensive and critical research paper, following points might be considered; 

Major Comments:

  • The body of article is well summarized and categorized. But it might be better if the reviewed studies come in form of table, outlining the research methods, outcomes and examined cognitive function domains.

Response: Done. We thank the reviewer for this suggestion. Three Tables have been added (page 40).

  • As it is mentioned in the body of article, one of the major reasons of having different results in previous studies, is the cognitive assessment methodology. Therefore, “Summary and conclusion” section should contain critical explanation about differences between cognitive assessments (e.g., MMSE, SPMSQ, WAIS-III &…) and its possible effects on results of study.

Response: Text has been added in response to this comment.

Modification to the text in the section 10: Previous studies have shown that different methodologies for assessing cognitive performance may contribute to equivocal results; future studies may investigate biomarkers and utilise neuroimaging techniques for screening MCI.

  • Since the section 8 is named as “Potential mechanisms”, some scientific mechanism of action must be outlined in this section. So far, there are number of studies which investigated the possible mechanism of Vitamin D, Inflammation and exercise on cognition and skeletal muscle.

Response: We have made an attempt to explain how vitamin D, inflammation and exercise might affect both muscle and brain function, thereby suggesting ways the two might be connected and a new section (8.3) has been added to the manuscript.

Vitamin D, exercise and inflammation

Vitamin D inhibits inflammation through regulating the production of inflammatory cytokines and inhibiting the proliferation of proinflammatory cells 105, which could be beneficial to skeletal muscle heath and brain health 106. A Iranian study found that vitamin D3 supplementation reduced the mRNA expression levels of IL-17A and IL-6, but increased the level of IL-10 107. This study focused on patients with multiple sclerosis.

Exercise through muscle contractions induces myokines (e.g. IL-6, brain-derived neurotrophic factor, BDNF) that affect lipid and glucose metabolism. Recovery from IL-6 peak following exercise can dampen inflammation and oxidative burst activity; the anti-inflammatory effect of BDNF may contribute to this recovery. It is well known that chronic exercise down-regulates systemic inflammation and is an effective management strategy for insulin resistance 108. Thus, exercise-induced down-regulation of IL-6 and the anti-inflammatory effect of BDNF may be related pathways. Thus, dietary, UV-associated or supplementary vitamin D combined with excise may effect both brain and skeletal muscle health. For example, a study in the USA suggested that supplemental vitamin D combined with intense exercise may enhance the recovery of muscle strength in adults who suffer a muscle damage event 109.

  1. “Summary and conclusion” section might better to contain total number of research articles, average length and other characteristics of studies which are reviewed in this stud

Response: We have modified the text accordingly.

Modification to the text in the section 10: “The main body of text in this review included and critically examined 28 key research articles in terms of cross-sectional studies, longitudinal studies and clinical trials that demonstrated relationships between skeletal muscle and cognitive function. (Table 1, 2, 3).”

Minor comments:

  • A comma is needed following the word “neck area” in Page 4, line 163;

“The study measured muscle volume in the neck area general cognitive function using MMSE, and cognitive domains of memory and executive function using Rey’s auditory-verbal declarative memory test and the controlled word association test, respectively.”

Response: Done. We thank the reviewer for this comment.

  1. pQCT is a test for examining bone density. Thus, for better understanding the text, it’s better to change the place of “bone” and “cognitive impairment” in the following sentence. Otherwise it might be misunderstood as a cognitive impairment test [Page 7, line 335];

“A UK study investigated bone and cognitive impairment using peripheral quantitative computed tomography (pQCT), and reported that gait speed, but not bone density/structure, was associated with MCI.”

Response: Done. We thank the reviewer for this comment.

Page 7, line 343; “dual-energy X-ray absorptiometry” is should be written as “DXA” which the abbreviation is mentioned before in Page 4, line 152.

Response: Done. We thank the reviewer for this comment.

Round 2

Reviewer 2 Report

Manuscript is well revised according to the comments.

It seems to be ready for acceptance. Great job done.